# Population-based study of eclampsia: Lessons learnt to improve maternity care

Diane Korb[1,2]*, Elie Azria[1,3], Priscille Sauvegrain[1,4], Lionel Carbillon[5], Bruno Langer[6], Aurélien Seco[1,7], Coralie Chiesa-Dubruille[1], Marie Hélène Bouvier-Colle[1], Epimoms study group[¶], Catherine Deneux-Tharaux[1]

1 CRESS, Obstetrical Perinatal and Pediatric Epidemiology Research Team, EPOPé, INSERM, INRA, Université Paris Cité, Paris, France, 2 Department of Obstetrics and Gynaecology, Robert Debré Hospital, AP-HP, Université de Paris, Paris, France, 3 Maternity Unit, Groupe Hospitalier Paris Saint Joseph, FHU PREMA, Université de Paris, Paris, France, 4 Department of Obstetrics and Gynaecology, Pitié-Salpêtrière Hospital, AP-HP, Sorbonne Université, Paris, France, 5 Department of Obstetrics and Gynaecology, Jean Verdier Hospital, AP-HP, Bondy, Sorbonne North Paris University, Paris, France, 6 Pôle de Gynécologie-Obstétrique, Hôpital de Hautepierre, Avenue Molière, Strasbourg, Université de Strasbourg, Strasbourg, France, 7 Clinical Research Unit Paris Centre, Assistance Publique des Hôpitaux de Paris, Paris, France

¶ Membership of the Epimoms study group is provided in the Acknowledgments
* diane.korb@inserm.fr

**Data Availability Statement:** Data cannot be shared publicly due to the small number of cases of eclampsia in our one year study period. According to French legislation and ethical approval, participants' anonymity must be

## Abstract

### Background

Among hypertensive disorders of pregnancy (HDP), eclampsia is a rare but serious event, often considered avoidable. Detailed assessment of the adequacy of care for the women who have eclampsia can help identify opportunities for improvement and for prevention of the associated adverse maternal and neonatal outcomes.

### Objective

1/ To estimate the incidence and describe the characteristics of women with eclampsia and to compare them with those of women with non-eclamptic hypertensive disorders of pregnancy (HDP)-related severe maternal morbidity (SMM) and of control women without SMM 2/ To analyse the quality of management in women who had eclampsia, at various stages of their care pathway.

### Methods

It was a planned ancillary analysis of the EPIMOMS population-based study, conducted in six French regions in 2012–2013. Among the 182,309 maternities of the source population, all women with eclampsia (n = 51), with non-eclamptic HDP-related SMM (n = 351) and a 2% representative sample of women without SMM (n = 3,651) were included. Main outcome was the quality of care for eclampsia assessed by an independent expert panel at three different stages of management: antenatal care, care for pre-eclampsia and care for eclampsia.

guaranteed. Data are available for researchers who meet the criteria by contacting INSERM EPOPé Institutional Data Access.

**Funding:** The authors received no specific funding for this work.

**Competing interests:** The authors have declared that no competing interests exist.

## Results

The eclampsia incidence was 2.8 per 10,000 (95%CI 2.0–4.0). Antenatal care was considered completely inadequate or substandard in 39% of women, as was pre-eclampsia care in 76%. Care for eclampsia was judged completely inadequate or substandard in 50% (21/42), mainly due to inadequate use of magnesium sulphate.

## Conclusion

The high proportion of inadequate quality of care underlines the need for an evidence-based standardisation of care for HDP.

## Introduction

Hypertensive disorders of pregnancy (HDP) are a major cause of severe maternal complications [1–3]. Among these, eclampsia is a rare but serious event that accounts for one-third to one-half of the hypertensive causes of maternal mortality—many considered avoidable [4]. Detailed assessment of the quality of care for the women who have eclampsia can help identify opportunities for improvement and potentially for prevention of the associated adverse maternal and neonatal outcomes. Previous studies have mainly focused on the management of the eclampsia event itself, in particular, the administration of magnesium sulphate in this context and showed there is room for improvement [5, 6]. However, a more comprehensive assessment of the global care trajectory of the women along the morbidity continuum, including antenatal care and the early screening and management of HDP would provide complementary insight into the overall quality of care and help identify means of prevention before eclampsia occurs. From this perspective, the analysis of women with a wider range of severe hypertensive complications, from women with severe pre-eclampsia who do not develop eclampsia to women with eclampsia, should yield more information on potential areas for better prevention.

The multiregional, population-based EPIMOMS study, specially designed to examine severe maternal morbidity (SMM), offers a unique opportunity to use detailed prospectively collected data to analyse eclampsia cases with concomitant cases of other HDP-related SMM [7].

The aims of this planned ancillary analysis were first to describe the characteristics of women with eclampsia and to compare them with those of women with non-eclamptic HDP-related SMM and of control women without SMM; and second, to analyse the adequacy of management in women who had eclampsia at different stages of their care pathway. These objectives were chosen to identify opportunities for primary and secondary prevention of eclampsia.

## Materials and methods

### Study population

The source population came from the EPIMOMS project, a prospective population-based study specially designed to investigate SMM and conducted in six French regions from May 1, 2012, through November 30, 2013 (recruitment and data collection).[7] Recruitment took place over a one-year period in each region at 119 maternity units and 136 intensive care units (regardless of the hospitals level of care, number of annual deliveries, or public or private

status) that accounted for 182,309 maternities during the study period, i.e., one fifth of those in France during that period, with characteristics of parturients and hospitals similar to the national profile [8]. In France, the organization of perinatal care is regionalized. Women choose the maternity unit where they plan to give birth; however, in case of complications occurring during pregnancy or if they are at high risk of obstetrical complication, they are referred to a maternity unit of the region with the appropriate level of care for them and their foetus.

The EPIMOMS study began by developing a multicriteria standardised definition of SMM, using a national Delphi formal expert consensus process intended to characterise maternal complications involving severe health impairment and organ dysfunction (criteria of the Epimoms SMM definition in S1 Table). Eclampsia was defined by the occurrence of seizures in a woman diagnosed with pre-eclampsia, or whose seizures could not be attributed to another cause, in accordance with the national guidelines on multidisciplinary management of severe pre-eclampsia [9–11].

All women who met one or more of the definition's criteria of an SMM event during pregnancy (after 22 weeks of gestation) or in the 42 days postpartum were prospectively identified and included in the EPIMOMS study (n = 2,540). The completeness of case identification was verified through the review of birth logbooks, hospital discharge data and laboratory files.

Concomitantly, as per the EPIMOMS study protocol, a 1/50 random sample of women who gave birth without SMM in the same regions during the same time period was selected as a control group.

For this analysis, conducted in 2021, we selected 3 study groups: women with eclampsia (n = 51), with each case of eclampsia reviewed to ensure the accuracy of the diagnosis; women with SMM related to a non-eclamptic HDP (n = 351), and the women without SMM in the EPIMOMS control group (n = 3,651).

## Data collection

Data collected concerned women's social (country of birth, sources of income, standard health insurance, if they lived alone) and demographic (maternal age, body mass index) characteristics, pre-existing chronic hypertension, previous gestational hypertensive disorders, and characteristics of the current pregnancy (in vitro fertilisation, multiple pregnancy). These data were collected by a manual review of medical records by research midwives trained for this study and entered in a specific electronic report form developed for the study, and were anonym. Additionally, for women with SMM, the criteria, causes, clinical timing (antepartum, intrapartum or postpartum) and details about the course and management of the severe morbid event were prospectively collected by the clinician in charge.

Specific data for women with eclampsia included the timing of the first seizure and detailed clinical and laboratory data at two time points. The first time point was just before eclampsia (when opportunities for primary prevention may have been present), and was defined, for eclampsia that occurred before birth, as the last antenatal visit or the last 48 hours of hospitalisation if the woman was admitted, and for eclampsia that occurred in the postpartum, as the post-birth hospital stay. The second time point was the eclampsia episode (when opportunities for secondary prevention may have been present). Data collected at these two time points included each patient's reported symptoms, physical examination findings (maximum systolic and diastolic blood pressure, functional signs of hypertensive disorders), laboratory values (hemoglobinaemia, platelets, hepatic and renal function) and management (antihypertensive treatment, magnesium sulphate administration prophylactic and during or immediately after the event), other anticonvulsant treatment, mode of birth and transfer to intensive care unit).

We also collected the occurrence of other severe maternal complications (maternal death, sub-capsular hepatic hematoma, placental abruption, HELLP syndrome, disseminated intravascular coagulopathy, severe obstetric hemorrhage, posterior reversible encephalopathy syndrome (PRES), or a stroke), as well as perinatal death.

## Assessment of adequacy of care in women with eclampsia

The quality of care in the group of women with eclampsia was assessed by a panel of 13 independent experts (4 anaesthesiologists, 5 obstetricians and 4 midwives). Based on an anonymised copy of all medical reports, they reviewed each case of eclampsia and reached a consensual decision on the quality of care during a 3-day session. The reference was the national recommendations on management of hypertension-related disorders [9]. Three different stages of management were analysed: antenatal care (before a diagnosis of pre-eclampsia), care for pre-eclampsia (if the woman was diagnosed with pre-eclampsia before eclampsia); and care during and after the eclampsia episode. For each stage of management, the experts assessed the quality of care in 5 categories: completely inadequate, substandard, fairly adequate, completely adequate, and impossible to judge (when informative data were missing in the medical charts). If the experts classified the management as completely inadequate or substandard, they had to determine whether the inadequate quality was related to the care content, the care organisation and/or the woman's adherence. Conclusions were captured via a standardised data collection form.

This detailed review and assessment of quality of care could be conducted for 44 of the 51 women with eclampsia. One region, Lorraine, which reported six women with eclampsia, did not participate in this additional data collection, and the medical file of one other woman was unavailable for this review.

In addition to the individual data collection, data about the general characteristics of the maternity hospitals and the existence and content of local protocols for the management of hypertensive disorders of pregnancy were collected in a specific questionnaire completed by the head of each participating maternity unit.

## Statistical analyses

The incidence of eclampsia was calculated as the number of women with eclampsia divided by the total number of women who gave birth in the participating maternity units during the inclusion period, with a binomial 95% confidence interval.

The socio-demographic characteristics of the women, their medical and obstetric histories, and the characteristics of their pregnancy were described for each group and compared: the women with eclampsia, those with SMM related to non-eclamptic HDP and the representative sample of women without SMM.

For women with eclampsia, we described the timing of eclampsia occurrence, the mode of and gestational age at birth, the clinical and laboratory characteristics during the period preceding eclampsia and during the eclampsia episode, the management of eclampsia, and the maternal and neonatal complications.

The quality of care in women with eclampsia, in the 5 categories described above, was reported for each of the three stages of management.

Proportions are presented as percentages and skewed distributions as medians with interquartile ranges (IQRs). Mean and Standard Deviation are presented for normal distributions. The comparisons of characteristics between groups were tested with Student tests, $Chi^2$, Fisher exact tests, as appropriate. All analyses were carried out with STATA v13 software (StataCorp, College Station, TX, USA).

### Ethics approval

The National Data Protection Authority (Commission Nationale de l'Informatique et des Libertés [CNIL] authorisation no. 912210, Mar. 14, 2012) approved the EPIMOMS study. The requirement for written informed consent was waived, in accordance with French legislation at that time, because all women received standard care and all data were anonymised.

## Results

### Incidence and women characteristics

During the study period, 182,309 women gave births in the maternity units of the 6 regions covered by the EPIMOMS study. Fifty-one cases of eclampsia occurred, for a population-based incidence of eclampsia of 2.8 per 10,000 (95% CI 2.0–4.0).

Women with eclampsia were more often born outside Europe, compared with both the women with SMM related to non-eclamptic HDP and women without SMM (Table 1); the proportion of women born in sub-Saharan Africa was particularly high in the eclampsia group (20.0% compared with 15.0% and 5.7% respectively). Women with eclampsia also more often had no source of legal work-related income and no standard health insurance and more often lived without a partner, although the difference with women with non-eclamptic HDP-related SMM did not reach statistically significance.

**Table 1. Maternal and obstetric characteristics in women with eclampsia, in women with non-eclamptic hypertensive disorders-related SMM and in women without SMM.**

| | Eclampsia | Non-eclamptic HDP-related SMM | p[a] | Controls | p[b] |
|---|---|---|---|---|---|
| | N = 51 | N = 351 | | N = 3651 | |
| **Characteristics** | **n (%)** | **n (%)** | | **n (%)** | |
| **Maternal characteristics (data available)** | | | | | |
| Maternal age (years) (mean±SD)(n = 51/351/3651) | 29.2±6.1 | 31.5±5.7 | 0.007 | 30.5±5.2 | 0.068 |
| <25 | 19 (37.3) | 47 (13.4) | 0.001 | 568 (15.6) | 0.001 |
| [25–29] | 10 (19.6) | 109 (31.1) | | 1142 (31.3) | |
| [30–34] | 12 (23.5) | 87 (24.8) | | 1205 (33.0) | |
| [35–39] | 8 (15.7) | 89 (25.4) | | 599 (16.4) | |
| ≥40 | 2 (3.9) | 19 (5.4) | | 137 (3.8) | |
| Body mass index (kg/m2) (n = 51/325/3417) | | | 0.821 | | 0.046 |
| <30 | 34 (79.1) | 252 (77.5) | | 3122 (88.8) | |
| ≥30 | 9 (20.9) | 73 (22.5) | | 395 (11.2) | |
| Region of birth (n = 50/327/3063) | | | 0.016 | | <0.001 |
| Europe | 25 (50.0) | 227 (70.7) | | 2424 (79.1) | |
| North Africa | 8 (16.0) | 28 (8.7) | | 316 (10.3) | |
| Sub-Saharan Africa | 10 (20.0) | 48 (15.0) | | 173 (5.7) | |
| Other | 7 (14.0) | 18 (5.6) | | 150 (4.9) | |
| Sources of income during pregnancy (n = 45/275/2163) | | | 0.002 | | <0.001 |
| Salary or other income from a professional activity | 19 (45.2) | 190 (69.6) | | 1784 (82.8) | |
| Other | 23 (54.8) | 83 (30.4) | | 372 (17.2) | |
| Standard health insurance (n = 48/284/2539) | | | 0.065 | | 0.003 |
| Yes | 36 (75.0) | 243 (85.6) | | 2254 (88.8) | |
| No | 12 (25.0) | 41 (14.4) | | 285 (11.2) | |
| Living without partner (n = 48/328/3380) | 6 (12.5) | 21 (6.4) | 0.126 | 135 (4.0) | 0.003 |
| Previous chronic hypertension (n = 51/347/3650) | 2 (3.9) | 34 (9.8) [†] | 0.172 | 30 (0.8) | 0.018 |

In addition, compared with women without SMM, women with eclampsia and those with non-eclamptic HDP-related SMM had a higher prevalence of classical risk factors for HDP: chronic hypertension, gestational hypertensive disorder in a previous pregnancy, and multiple pregnancies.

## Eclampsia event

Eclampsia occurred in the antepartum period in 23 (45.1%) women, intrapartum in 8 (15.7%) women, and postpartum in 20 (39.2%) women (Table 2, S1 Fig). It was inaugural, ie, without a diagnosis of pre-eclampsia before the eclampsia event, for 18/51 (35.3%) women.

Table 2 presents in the group of women with eclampsia, clinical alert symptoms and missed opportunities in management before the eclamptic event. Moreover, among women who had a maximum systolic blood pressure ≥160 mmHg or a maximum diastolic blood pressure ≥110 mmHg before eclampsia, 7/16 (43.8%) did not receive antihypertensive treatment. Among the 19/49 (38.8%) women with premonitory symptoms, 3/19 (15.8%) were not hospitalised.

Among the 33/51 (64.7%) women with pre-eclampsia diagnosed before the eclampsia, 17 (51.5%) did not receive antihypertensive treatment, 14 (42.4%) were not hospitalised, 14 (42.4%) had a maximum systolic blood pressure ≥160 mmHg, 6 (18.2%) a maximum diastolic blood pressure ≥110 mmHg, and 14 (42.4%) had premonitory symptoms. Only one woman received magnesium sulphate prophylaxis, administered 45 minutes before the convulsions.

Among the 18 women who did not have pre-eclampsia diagnosed before eclampsia, at the last measurement before eclampsia, 2 (11.1%) had a maximum systolic blood pressure ≥160

**Table 2. Characteristics and management of women in the eclampsia group before the eclamptic event.**

| | | | | | | | Eclampsia |
|---|---|---|---|---|---|---|---|
| | | | | | | | **n = 51** |
| **Characteristics before eclampsia** | | | | | | | **n (%)** |
| Maximum blood pressure ≥ 160/110 mmHg | | | | | | | 16/51 (64.7%) |
| | If yes (n = 16): | Antihypertensive treatment | | | | | 9 (56%) |
| | | Lack of antihypertensive treatment | | | | | 7 (44%) |
| Premonitory symptoms[a] | | | | | | | 19/49 (38.8%) |
| | If yes (n = 19): | Hospitalisation | | | | | 16 (84%) |
| | | Lack of hospitalisation | | | | | 3 (16%) |
| Pre-eclampsia diagnosed before eclampsia | | | | | | | 33/51 (65%) |
| | If yes (n = 33): | Lack of antihypertensive treatment | | | | | 17 (52%) |
| | | Lack of hospitalisation | | | | | 14 (42%) |
| | | Maximum systolic blood pressure ≥160 mmHg | | | | | 14 (42%) |
| | | Maximum diastolic blood pressure ≥110 mmHg | | | | | 6 (18%) |
| | | Premonitory symptoms | | | | | 14 (42%) |
| | | Magnesium sulphate prophylaxis[b] | | | | | 1 (3%) |
| | If no (n = 18): | Maximum | | | | | 2 (11%) |
| | | sSystolic blood pressure ≥160 mmHg | | | | | |
| | | Maximum diastolic blood pressure ≥110 mmHg | | | | | 0 |
| | | Premonitory symptoms | | | | | 5 (28%) |
| | | Magnesium sulphate prophylaxis | | | | | 0 |

[a]: headaches, visual disturbance, epigastric pain, nausea, vomiting, edema, hyper-reflexia, tinnitus, other
[b]: magnesium sulphate prophylaxis administered 45 minutes before the convulsions

mmHg, none had a maximum diastolic blood pressure ≥110 mmHg and 5 (27.8%) had pre-monitory symptoms. None of these women received magnesium sulphate prophylaxis.

Table 3 presents the main clinical and biological characteristics and the management of eclampsia cases.

During the eclamptic event, magnesium sulphate treatment was administered to 42/51 (82.4%) women. No maternal deaths and no strokes occurred. Three (5.9%) women had a severe postpartum haemorrhage, nine (17.6%) HELLP syndrome and six (11.8%) women a PRES. Five (9.1%) women experienced the perinatal death of the fetus/neonate. (Table 3).

## Assessment of adequacy of care in women with eclampsia

Table 4 and Fig 1 describe the experts' consensual assessments of the quality of care in women with eclampsia. Antenatal care before HDP was considered completely inadequate or substandard for 39% of women (15/38). This was mainly due to insufficient clinical surveillance: visits too far apart, inadequate monitoring of proteinuria and of blood pressure, in particular, when subnormal results were found. Lack of adherence to prescribed care was reported for two women.

Care for pre-eclampsia was considered completely inadequate or substandard for 76% (19/25 of the women with pre-eclampsia diagnosed before eclampsia). This was mainly due to late or inadequate antihypertensive drugs and magnesium sulphate administration, and insufficient surveillance of blood parameters. Lack of adherence was noted for one woman.

Care for eclampsia was considered completely inadequate or substandard for 50% (21/42). This was mainly due to inadequate use of magnesium sulphate—either not used (5/42, 11.9%) or used at an inadequate dosage or poorly timed (13/42, 31.0%)—and to inadequate monitoring after the eclamptic episode—too short and in an inappropriate place. No lack of adherence was noted.

Most the cases deemed "impossible to judge" because of insufficient reported information in the medical files involved women born in sub-Saharan Africa: 4/6 for antenatal care, 2/2 for pre-eclampsia and 1/2 for eclampsia.

Among the 351 women with non-eclamptic HDP-related SMM, 189 (53.8%) had reported premonitory symptoms, and 31 (16.4%) of them received magnesium sulphate prophylaxis.

## Protocols for management of preeclampsia and eclampsia

aThe existence of protocols for management of preeclampsia and eclampsia among the participating maternity units and the content of these protocols are reported in Table 5. Among the 119 maternity units participating in EPIMOMS, 30/112 (27%) did not have a protocol for anti-hypertensive drug use during pregnancy. There was no protocol for management of pre-eclampsia in 20/113 (18%) units, and none for eclampsia management in 21/108 (19%) units. Prophylactic administration of magnesium sulphate in pre-eclampsia was not recommended in 26/112 units (23%).

## Discussion

In this population-based study 1 out of every 3,600 women who gave birth had eclampsia. Socially vulnerable subgroups were more represented among women with eclampsia than among those with non-eclamptic HDP-related SMM or control women, in particular migrant women from sub-Saharan Africa. The care provided to women with eclampsia was completely inadequate or substandard for the majority of them and suggests major opportunities for improvement at two main stages: in the management of pre-eclampsia before eclampsia and

**Table 3. Clinical and biological features and management of the eclampsia event.**

|  |  | Eclampsia |
|---|---|---|
|  |  | N = 51 |
| **Characteristics** |  | **n (%)** |
| **Timing of first seizure (n = 51)** |  |  |
|  | Antepartum | 23 (45.1) |
|  | Intrapartum | 8 (15.7) |
|  | Postpartum | 20 (39.2) |
| **Gestational age at birth (weeks)** |  |  |
|  | <28 | 3 (5.9) |
|  | [28–32] | 7 (13.7) |
|  | [33–36] | 17 (33.4) |
|  | $\geq$ 37 | 24 (48.0) |
| **Clinical signs during eclampsia event (n (%)) (n = 51)** |  |  |
|  | Inaugural[a] eclampsia | 18 (35.3) |
|  | Maximum systolic blood pressure $\geq$ 160 mmHg (n = 43) | 36 (83.7) |
|  | Maximum diastolic blood pressure $\geq$ 110 mmHg (n = 43) | 20 (46.5) |
|  | Seizure | 51 (100.0) |
|  | Place of occurrence (n = 49) |  |
|  | Hospital | 38 (77.6) |
|  | Home | 9 (18.4) |
|  | Other | 2 (4.0) |
|  | Recurrent seizures (n = 48) | 22 (45.8) |
|  | Recurrent seizures = 2 | 16 |
|  | Recurrent seizures $\geq$ 3 | 6 |
|  | Eclampsia without high blood pressure in pre or postpartum | 0 |
| **Biology** |  |  |
|  | Hemoglobinaemia< 10.5 g/dL (n = 46) | 29 (63.0) |
|  | Platelets<100 103/mm3 (n = 46) | 19 (41.3) |
|  | Creatininaemia>135 (n = 39) | 3 (7.7) |
|  | AST> 2N (n = 40) | 20 (50.0) |
|  | ALT> 2N (n = 41) | 11 (26.8) |
| **Management (n = 51)** |  |  |
|  | Antihypertensive treatment | 44 (86.3) |
|  | Magnesium sulphate | 42 (82.4) |
|  | Dose (med (Q1-Q3), grams) | 4 (2–11) |
|  | Clinical signs of overdose | 0 (0.0) |
|  | Time interval between eclampsia and sulphate (med (Q1-Q3), min) | 15 (3–50) |
|  | Benzodiazepine | 11 (21.6) |
|  | Other anticonvulsant | 11 (21.6) |
| **Mode of birth (n = 51)** |  |  |
|  | Vaginal birth[b] | 15 (29.4) |
|  | Cesarean during labor[c] | 10 (19.6) |
|  | Cesarean before labor[d] | 26 (51.0) |
| **Time interval between (med (Q1-Q3), min)[e]** |  |  |
| Ante or intrapartum eclampsia and birth |  |  |
| Cesarean birth (n = 29) |  | 64 (32–159) |
| Vaginal birth (n = 2) |  | 16 (12–44) |
| Birth and postpartum eclampsia |  |  |

*(Continued)*

**Table 3.** (Continued)

| Characteristics | | Eclampsia |
|---|---|---|
| | | **N = 51** |
| **Characteristics** | | **n (%)** |
| Cesarean birth (n = 7) | | 239 (112–428) |
| Vaginal birth (n = 10) | | 343 (197–620) |
| **Transfer to intensive care unit (n = 51)** | | 34 (66.7) |
| **Maternal complications (n = 51)** | | |
| | Maternal death | 0 |
| | Severe obstetric hemorrhage | 3 (5.9) |
| | Subcapsular hepatic hematoma | 0 |
| | Placental abruption | 0 |
| | HELLP syndrome | 9 (17.6) |
| | Disseminated intravascular coagulopathy | 0 |
| | PRES syndrome | 6 (11.8) |
| | Cerebrovascular accident | 0 |
| **Perinatal outcomes** | | |
| | Small-for-gestational age at birth[f] (n = 54) | 23 (42.6) |
| | Perinatal death (n = 55) | 5 (9.1) |
| | Fetal death | 3 (5.5)[g] |
| | Intrapartum death | 1 (1.8)[h] |
| | Early neonatal death (<7 days) | 1 (1.8)[i] |

[a] Inaugural: without a diagnosis of pre-eclampsia before eclamptic event

[b] 1/15 eclampsia intrapartum; 14/15 eclampsia postpartum

[c] 6/10 eclampsia ante/intrapartum

[d] 22/26 eclampsia antepartum; 23/26 cesarean for hypertensive complications

[e] exclusion of 2 cases with postpartum eclampsia after discharge (D5, D11)

[f] defined by the <5th percentile according to EPOPé curves

[g] severe fetal growth restriction

[h] 25 weeks of gestation, induction for severe pre-eclampsia, 400 g

[i] 23 weeks of gestation, cesarean for severe pre-eclampsia, 350 g

in the management of the eclamptic episode. In particular, we found an inappropriately low rate of magnesium sulphate use for both primary and secondary prevention of eclampsia.

In this first population-based estimation of the incidence of eclampsia in France, we found a 1/3,600 rate, in the range of those reported in other high-income countries such as the UK, Australia and New Zealand and Italy, with similar or close definitions [6, 10, 11]. A lower rate has recently been reported in the Netherlands after an active campaign to improve care [12].

We found that socially vulnerable women, in particular, migrant women from sub-Saharan Africa, accounted for a larger portion of the group of women with eclampsia than the women with non-eclampsia HDP-related SMM and controls. One potential explanation for this finding could be a specific phenotype of the disease in these women with acute inaugural onset of eclampsia not preceded by the prior continuum of milder gestational hypertensive disorders. Another explanation supported by previous qualitative and quantitative data could be that their unfavourable social context constitutes a barrier to access timely and adequate care for early signs, which would increase the risk of more severe and morbid complications due to not medically justified differential care [13, 14]. The fact that these sub-Saharan Africa women were over-represented among the eclampsia cases classified as "impossible to judge" is a

**Table 4. Reasons for inadequate care[a].**

| | Cases of eclampsia reviewed |
|---|---|
| | n/N (%) |
| **Inadequate antenatal care** | **15/38 (39)** |
| **Inadequate content of care** | **12 (80)** |
| Insufficient screening | 10 |
| *Visits too far apart, no re-evaluation by ultrasound* | *6* |
| *Inadequate surveillance of blood pressure* | *2* |
| Inadequate treatment of hypertension | 3 |
| **Inadequate organisation of care** | **2 (13)** |
| **Inadequate adherence by the woman** | **2 (13)** |
| Inadequate care for pre-eclampsia* | 19/25 (76) |
| **Inadequate content of care** | **19 (100)** |
| Insufficient surveillance of blood parameters | 11 (58) |
| *Antepartum* | *3* |
| *Intrapartum* | *5* |
| *After birth* | *6* |
| Inadequate treatment | 15 (79) |
| *Antihypertensive treatment late or inadequate* | *10* |
| *Delayed decision to induce birth* | *4* |
| *Magnesium sulphate treatment absent or insufficient* | *4* |
| **Inadequate organisation of care** | **3 (16)** |
| **Inadequate adherence by the woman** | **1 (5)** |
| Inadequate care for eclampsia | 21/42 (50) |
| **Inadequate care content** | **21 (100)** |
| Diagnosis not done | 5 (24) |
| Inadequate treatment (delayed or inappropriate) | 20 (95) |
| *Antihypertensive* | *3* |
| *Anticonvulsant* | *4* |
| *Magnesium sulphate* | *18* |
| *Not administered* | *5* |
| *Inadequate dosage and/or timing* | *13** |
| Insufficient surveillance | 10 (48) |
| *In antepartum* | *1* |
| *In intrapartum* | *1* |
| *After birth* | *9* |
| **Inadequate organisation of care** | **5 (24)** |
| **Inadequate adherence by the woman** | **1 (1)** |

Cases considered as "impossible to judge" were excluded

[a] "inadequate care" in this table combines « completely inadequate » and « substandard » care

* assessment among women with pre-eclampsia before eclampsia

** 6 women received an initial dose of MgSO4 lower than the recommended 4 grams, 7 women received only the initial dose without further perfusion of MgSO4, one woman received MgSO4 26 min after the seizures, 5 women received MgSO4 more than 3 hours after the seizures.

supplementary argument for a lower quality of care for these women, as the traceability of medical data in medical files is also a component of quality of care.

The two complementary approaches used in this study for the quality of care assessment both showed that 50% of the women received inadequate care during the eclampsia event, a

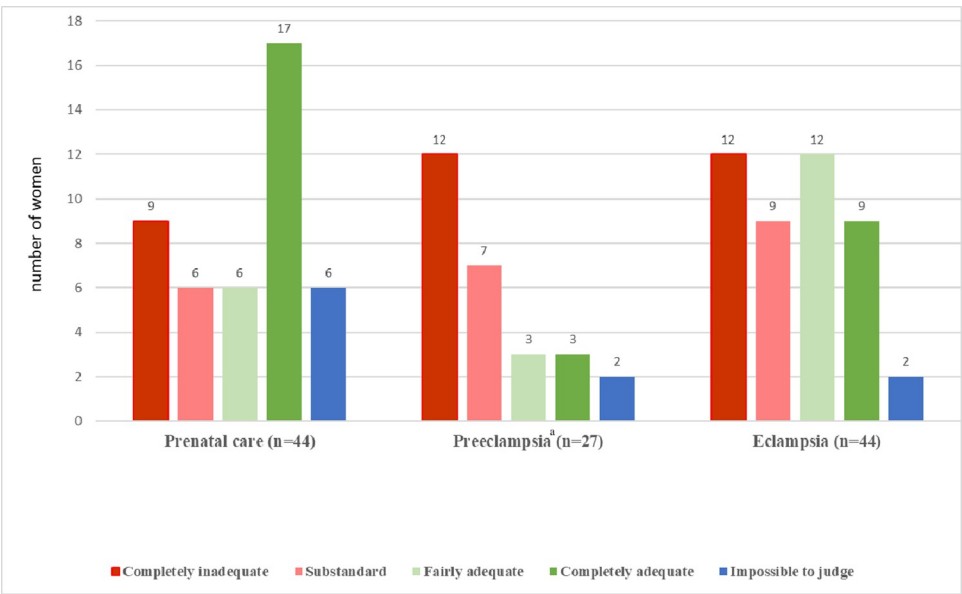

<sup>a</sup> Care for pre-eclampsia in the 27 women who were diagnosed with pre-eclampsia before eclampsia

**Fig 1. Adequacy of care in women with eclampsia assessed by the panel of experts.**

finding also reported in previous studies [6, 10]. In an original finding, we also identified failure in preventive care before eclampsia. We found, particularly in women diagnosed with preeclampsia, a lack of screening or underestimation of clinical and laboratory warning signs, which were present, and a failure to apply the recommended management when these signs were identified. This delay in appropriate care may be a potential mechanism explaining the evolution toward severe hypertensive complications and a worsening of maternal health. Of note, maternal adherence was rarely involved in the inappropriateness of care. This inadequacy of care provided may be a consequence of either a lack of knowledge of the recommendations or a failure to translate them into practice. It reveals the failure of the strategy for guidelines dissemination and appropriation. The notable proportion of maternity units without active risk management that have not translated the national recommendations into local protocols suggests a first possible area of improvement at the local level.

**Table 5. Protocols for diagnosis and management of preeclampsia and eclampsia among the participating maternity units (n = 119).**

| Existence of local protocol on: | | n (%) |
|---|---|---|
| Antihypertensive drug use during pregnancy | | |
| | Yes | 82 (73%) |
| | No | 30 (27%) |
| | Missing data | 7 (6%) |
| Management of pre-eclampsia | | |
| | Yes | 93 (82%) |
| | No | 20 (18%) |
| | Missing data | 6 (5%) |
| Management of eclampsia | | |
| | Yes | 87 (81%) |
| | No | 21 (19%) |
| | Missing data | 1 (1%) |

Magnesium sulphate is the recognised drug of choice to prevent the recurrence of seizures in women with eclampsia, in contrast with its prophylactic use in prevention of the first seizure, which is less consensual [15, 16]. We found that magnesium sulphate remains insufficiently administered during the eclamptic episode. The 82% rate of use of magnesium sulphate in our study population appears lower than the reported 87% in Finland, 89% in Italy, 95% in the Netherlands, and 99% rate in the UK [5, 10–12]. Beyond this rate, the assessment by the experts also highlighted that even when administered, the length and mainly the dose of administration were often not in the recommended ranges, in particular with an under-dosing of magnesium sulphate. These findings identify further areas for improvement, starting with the implementation of standardised protocols in all maternity units with a special focus on awareness of warning signs and on the indications and modalities of magnesium sulphate use.

The EPIMOMS standardised definition of eclampsia developed through a national Delphi consensus process and its large population-based design guarantee an accurate identification of cases and estimation of the incidence of this rare event. This design was also especially relevant for the analysis of the quality of care as it included the broad range of practices existing in real-life care settings. In addition to women with eclampsia, two groups of women were studied as comparators, in particular, the group of women with non-eclamptic HDP-related SMM. Although a complete exploration of the morbidity spectrum of HDP would also have included women with non-severe pre-eclampsia (not available with the EPIMOMS design), this was an informative aspect of our analysis. Another originality of this study was the assessment of the quality of care according to clinical guidelines in women with eclampsia by two complementary approaches: on the one hand, a quantitative description of management items and, on the other hand, a qualitative judgment by a panel of experts.

Because of the rarity of the event, the number of eclampsia cases limited the analysis of quality of care for some subgroups of women with reported high-risk profiles, notably women born in sub-Saharan Africa. The EPIMOMS study was conducted 10 years ago, and practices may have changed. However, although the national guidelines for pre-eclampsia management were updated in 2021 [16], we believe the analysis of the quality of care assessed from the EPIMOMS data, and the opportunities for improvement identified, are still relevant. Indeed, the recent guidelines are similar to the previous ones for the components of care assessed in our analysis, except for the prophylactic use of magnesium sulphate [10], which is now recommended for even broader clinical symptoms not restricted to neurological signs, as in the most recent international guidelines [17–22]. Future studies should be conducted to assess if professionals modify their practices to adopt these new recommendations. This is particularly expected since the process of guidelines development has recently been modified, in order to increase the quality of the overall process, to deliver more focused and easily assimilated recommendations, and to facilitate professional decision making, with an emphasis placed on dissemination [23].

This population-based study of women with eclampsia underlines the need to improve the quality of care provided for HDP at all stages of the morbidity continuum and for all women. One priority is to increase the translation of clinical guidelines into local protocols and their integration into individual practice. Future research with mixed approaches aiming to understand individual and environmental barriers in this integration process will help improve management and hopefully decrease maternal morbidity.

## Supporting information

**S1 Checklist. STROBE statement—checklist of items that should be included in reports of observational studies.**
(DOCX)

**S2 Checklist.** *PLOS ONE* **clinical studies checklist.**
(DOCX)

**S1 Table. EPIMOMS multicriteria standardised definition of severe acute maternal morbidity, developed through a national Delphi formal expert consensus process.**
(DOCX)

**S1 Fig. Gestational age at the time of the eclamptic episode (for women with antepartum or intrapartum seizures) or at delivery (for women with postpartum seizures).**
(DOCX)

## Acknowledgments

The authors thank the experts for their contribution to the assessment of the quality of care: anesthesiologists: Marie-Pierre Bonnet, Martine Bonnin, Hawa Keita, Brigitte Storme-Roelens; obstetricians: Gaël Beucher, Pierre-François Ceccaldi, Catherine Crenn-Hebert, Gilles Kayem, Camille Leray; midwives: Nathalie Baunot, Anne Chantry, Véronique Tessier.

The authors thank the coordinators of the participating regional perinatal networks: Alsace, Aurore, Auvergne, Basse-Normandie, MYPA, NEF, Paris Nord, 92 Nord, Lorraine; Chloé Barasinski, Sophie Bedel, Aline Clin D'Amour, Laurent Gaucher, Isabelle Le Creff, Blandine Masson Carole Ramousset, Mathias Rossignol, Zelda Stewart, Dalila Talaourar, Yacine Toure, and Nicole Wirth for their contribution to the implementation of the EPIMOMS study in their region; the obstetricians, midwives and anesthetists who contributed to case identification and documentation in their hospital; and the research assistants who collected the data.

The authors also thank Jo Ann Cahn for editorial assistance.

## EPIMOMS Study Group

Elie Azria[1,2], Nathalie Baunot[2], Gaël Beucher[3], Marie-Pierre Bonnet[1], Marie-Hélène Bouvier-Colle[1], Lionel Carbillon[4], Anne Chantry[1], Coralie Chiesa-Dubruille[1], Catherine Crenn-Hebert[5], Catherine Deneux-Tharaux[1], Corinne Dupont[6], Jeanne Fresson[7], Gilles Kayem[1], Bruno Langer[8], Alexandre Mignon[9] Patrick Rozenberg[10], René-Charles Rudigoz[6], Aurélien Seco[1], Sandrine Touzet[11], Françoise Vendittelli[12], , , , , , , , , , , ,

Lead author: Catherine Deneux-Tharaux; catherine.deneux-tharaux@inserm.fr

[1] Université Paris Cité, CRESS, Obstetrical Perinatal and Pediatric Epidemiology Research Team, EPOPé, INSERM, INRA,, F-75014, Paris, France.

[2] Paris Nord Perinatal Network

[3] Basse-Normandie Perinatal Network

[4] Department of Obstetrics and Gynaecology, Jean Verdier Hospital, AP-HP, Bondy, Sorbonne North Paris University, France.[1]

[5] Paris Nord Perinatal Network

[6] Rhône-Alpes Aurore Perinatal Network

[7] Lorraine Perinatal Network

[8] Pôle de Gynécologie-Obstétrique, Hôpital de Hautepierre, Avenue Molière, 67098 Strasbourg, Université de Strasbourg, France

[9] Société Française d'Anesthésie Réanimation

[10] MYPA Perinatal Network, Ile de France region

[11] Rhône-Alpes Aurore Perinatal Network

[12] Auvergne Perinatal Network

## Author Contributions

**Conceptualization:** Diane Korb, Elie Azria, Catherine Deneux-Tharaux.

**Formal analysis:** Diane Korb, Aurélien Seco, Catherine Deneux-Tharaux.

**Methodology:** Diane Korb, Elie Azria, Priscille Sauvegrain, Aurélien Seco, Catherine Deneux-Tharaux.

**Project administration:** Coralie Chiesa-Dubruille, Marie Hélène Bouvier-Colle.

**Validation:** Diane Korb.

**Writing – original draft:** Diane Korb, Elie Azria, Catherine Deneux-Tharaux.

**Writing – review & editing:** Diane Korb, Priscille Sauvegrain, Lionel Carbillon, Bruno Langer, Aurélien Seco, Coralie Chiesa-Dubruille, Marie Hélène Bouvier-Colle.

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
