## [Decision Letter · Decision Letter 0]

18 Dec 2023

PONE-D-23-10835Population-based study of eclampsia: lessons learnt to improve maternity carePLOS ONE

Dear Dr. Korb

Thank you for submitting your manuscript to PLOS ONE. After careful consideration, we feel that it has merit but does not fully meet PLOS ONE’s publication criteria as it currently stands. Therefore, we invite you to submit a revised version of the manuscript that addresses the points raised during the review process. This is an important topic – is the standard of care met when caring for women with severe hypertensive disorders and eclampsia? As this is a rare event, clinicians are likely to be less familiar with it, and thus less likely to adhere to protocols and best practice. This was a planned ancillary analysis, of a large study, and so adds value.

We look forward to receiving your revised manuscript.

Kind regards,

Natasha L Pritchard

Academic Editor

PLOS ONE

Journal Requirements:

2. One of the noted authors is a group or consortium [Epimoms study group]. In addition to naming the author group, please list the individual authors and affiliations within this group in the acknowledgments section of your manuscript. Please also indicate clearly a lead author for this group along with a contact email address.

3. Please include a caption for figure 1.

Additional Editor Comments (if provided):

Dear authors

In addition to the comments by the peer reviewers, I have a few comments:

1) Although half of the women with an eclamptic event did not receive magnesium sulfate during their episode, I am aware the episodes can be so quick that there may not have been time for administration. In those that did not receive it during the episode, how many received it immediately after, as further seizure prophylaxis? This should be noted or discussed.

2) For the elevated BP >160/110, were these isolated readings, or more than once? If isolated, it may explain the lack of antihypertensive treatments as there may have been the presumption it should be retested.

3) I don’t believe the data supports hypotheses of a genetic predisposition of certain ethnic groups to altered preeclampsia and eclampsia presentations. I think this should be removed, and the conclusions based on the data.

4) Further acknowledgement in the discussion should occur that the timing may have been so acute there was no time to initiate management. However, it still adds value in assessing how far below protocol it frequently was.

5) Can it be clearer exactly what "optimal standard of care" was? 

Overall, it was a well written paper.

Reviewers' comments:

Reviewer's Responses to Questions

**Comments to the Author**

1. Is the manuscript technically sound, and do the data support the conclusions?

Reviewer #1: Yes

Reviewer #2: Partly

2. Has the statistical analysis been performed appropriately and rigorously? 

Reviewer #1: I Don't Know

Reviewer #2: Yes

3. Have the authors made all data underlying the findings in their manuscript fully available?

Reviewer #1: Yes

Reviewer #2: Yes

4. Is the manuscript presented in an intelligible fashion and written in standard English?

Reviewer #1: Yes

Reviewer #2: Yes

5. Review Comments to the Author

Reviewer #1: Dear authors,

The eclampsia is a very important topic actually. This article doesn't(t present very important results but this is an ancillary study.

Abstract

1. Add background section

Material and methods

2. Report this section according to STROBE

Results

3. Add sections

Reviewer #2: The manuscript is sound, however, some conclusions lack evidence. The authors statement suggesting that specific genetic background could be causing the higher incidence of Women with Sub Saharan African background in the eclamptic group is less likely. The results provided suggest an association between lower socio-economic groups i.e. migrant populations.

As the guidelines being referenced are ~10 years old and the analysis is retrospective it does not give an accurate representation of management of pre-eclampsia and eclampsia currently.

Case-control study would be required to analyse difference in timing and dosing in the prevention of eclampsia in women with pre-eclampsia. The retrospective analysis does not allow for such conclusions.

6. PLOS authors have the option to publish the peer review history of their article (what does this mean?). If published, this will include your full peer review and any attached files.

Reviewer #1: No

Reviewer #2: No

---

## [Author Response · Author response to Decision Letter 0]

1 Feb 2024

Dear Editor, 

Thank you for your response on December 18th 2023, concerning our manuscript PONE-D-23-10835, entitled:” Population-based study of eclampsia: lessons learnt to improve maternity care” informing us you would be willing to give further consideration to a revised version. 

The authors are very grateful to the Reviewers and Editors for their constructive help. We think the paper has been much improved. Our revised version has taken into account all the following points raised by the Reviewers and Editors. 

All the authors have read and approved the revised version of the paper.

We hope our manuscript now meets the standards of Plos One.

Yours sincerely,

Diane Korb

 

We have checked this. 

2. One of the noted authors is a group or consortium [Epimoms study group]. In addition to naming the author group, please list the individual authors and affiliations within this group in the acknowledgments section of your manuscript. Please also indicate clearly a lead author for this group along with a contact email address.

We clarified the Epimoms study group with the list of the individual authors and their affiliations, and the lead author for this group.

3. Please include a caption for figure 1.

We included the caption.

We included captions for Supporting Information files at the end of the manuscript, and updated any in-text citations to match accordingly.

We reviewed all the references.

 

Editor

1. Although half of the women with an eclamptic event did not receive magnesium sulfate during their episode, I am aware the episodes can be so quick that there may not have been time for administration. In those that did not receive it during the episode, how many received it immediately after, as further seizure prophylaxis? This should be noted or discussed.

We agree with the editor that the eclamptic event can be very rapid and that magnesium sulfate can be administered during or just after the convulsions. This is what we have considered: the administration of magnesium sulfate during the eclamptic episode or just after. We have added this clarification to the 140-141: “magnesium sulphate administration during or immediately after the event”.

We did not report this result written by the Editor:” half of the women with an eclamptic event did not receive magnesium sulfate during their episode”. Indeed: in our population, as described in the manuscript on lines 253-254: “During the eclamptic event, magnesium sulphate treatment was administered to 42/51 (82.4%) women.” This result is also shown in Table 3.

Maybe the confusion stems from the analysis of the assessment of adequacy of care in women with eclampsia by the panel, which showed that “Care for eclampsia was considered completely inadequate or substandard for 50% (21/42).” - Lines 273 to 277. Different reasons explained this result, “This was mainly due to inadequate use of magnesium sulphate — either not used (5/42, 11.9%) or used at an inadequate dosage or poorly timed (13/42, 31.0%)”. These results are also in the table 4: only 5/42 women did not receive magnesium sulfate. 

2) For the elevated BP >160/110, were these isolated readings, or more than once? If isolated, it may explain the lack of antihypertensive treatments as there may have been the presumption it should be retested.

As explained in the Methods, in lines 137-138: we analyzed the maximum systolic and diastolic blood pressure. We added this precision of the “maximum” of the value in the tables 2 and 3, and in the results in lines 227, 233-234 and 238-239.

We agree with the Editor that it is a limitation to consider only one blood pressure value, but we did not collect all the measurements. Nevertheless, the blood pressure value is a rare data that is not often presented in published articles.

3) I don’t believe the data supports hypotheses of a genetic predisposition of certain ethnic groups to altered preeclampsia and eclampsia presentations. I think this should be removed, and the conclusions based on the data.

We agree with the Editor and have modified the sentence in the following lines 324-327: “One potential explanation for this finding could be a specific phenotype of the disease in these women with acute inaugural onset of eclampsia not preceded by the prior continuum of milder gestational hypertensive disorders.”

4) Further acknowledgement in the discussion should occur that the timing may have been so acute there was no time to initiate management. However, it still adds value in assessing how far below protocol it frequently was.

We agree with the Editor, as in his first question. We have taken into account the administration of magnesium sulfate during the eclampsia episode or just after the episode and we have modified the methods to clarify this point in line 140-141. 

5) Can it be clearer exactly what "optimal standard of care" was? 

We have not used this term "optimal standard of care" in the manuscript, tables or figures. As explained in line 149 to 167, the assessment of adequacy of care in women with eclampsia was carried out by a panel of multidisciplinary and independent experts on the basis of the national recommendations on management of pregnancy-related hypertensive disorders published by 4 national scientific societies : French Society of Anesthesia and Intensive Care (Société française d'anesthésie et de réanimation - Sfar); French College of Gynecologists and Obstetricians (Collège national des gynécologues et obstétriciens français - CNGOF); French Society of Perinatal Medicine (Société française de médecine périnatale - SFMP); French Society of Neonatology (Société française de néonatalogie - SFNN) (Pottecher T, Luton D, Zupan V, Collet M. [Multidisciplinary management of severe pre-eclampsia]. J Gynecol Obstet Biol Reprod (Paris). 2009 Jun;38(4):351–7). Three different stages of management were analysed: antenatal care (before a diagnosis of pre-eclampsia), care for pre-eclampsia (if the woman was diagnosed with pre-eclampsia before eclampsia); and care during and after the eclampsia episode. For each stage of management, the experts assessed the quality of care in 5 categories: completely inadequate, substandard, fairly adequate, completely adequate, and impossible to judge (when informative data were missing in the medical charts). If the experts classified the management as completely inadequate or substandard, they had to determine whether the inadequate quality was related to the care content, the care organisation and/or the woman's adherence.

Overall, it was a well written paper.

Reviewers' comments:

Reviewer's Responses to Questions

Reviewer #1: Dear authors,

The eclampsia is a very important topic actually. This article doesn't(t present very important results but this is an ancillary study.

Abstract

1. Add background section

We added it 

Material and methods

2. Report this section according to STROBE

We added sections

Results

3. Add sections

We added sections

Reviewer #2: 

1. The manuscript is sound, however, some conclusions lack evidence. The authors statement suggesting that specific genetic background could be causing the higher incidence of Women with Sub Saharan African background in the eclamptic group is less likely. The results provided suggest an association between lower socio-economic groups i.e. migrant populations.

We agreed with the Reviewer and we modified the sentence in lines 324-327: “One potential explanation for this finding could be a specific phenotype of the disease in these women with acute inaugural onset of eclampsia not preceded by the prior continuum of milder gestational hypertensive disorders.”

2. As the guidelines being referenced are ~10 years old and the analysis is retrospective it does not give an accurate representation of management of pre-eclampsia and eclampsia currently.

We agree with the reviewer and this limitation is acknowledged and discussed line 381 to 389. Indeed, the EPIMOMS study was conducted 10 years ago, and practices may have changed. However, although the national guidelines for pre-eclampsia management were updated in 2021, we believe the analysis of the quality of care assessed from the EPIMOMS data, and the opportunities for improvement identified, are still relevant. Indeed, the recent guidelines are similar to the previous ones for the components of care assessed in our analysis, except for the prophylactic use of magnesium sulphate, which is now recommended for even broader clinical symptoms not restricted to neurological signs, as in the most recent international guidelines.

In addition, our results constitute a baseline assessment for future research exploring changes in eclampsia management practices.

3. Case-control study would be required to analyse difference in timing and dosing in the prevention of eclampsia in women with pre-eclampsia. The retrospective analysis does not allow for such conclusions.

In this study, we described the administration of magnesium sulfate in women with severe pre-eclampsia and in women with eclampsia preceded or not by a pre-eclampsia. The objective of this study has not to test the differences if timing and dosing in the prevention of eclampsia could modify the occurrence of eclampsia. The objective of this study was to describe the management and care of women with eclampsia.

---

## [Editor Report · Decision Letter 1]

27 Mar 2024

Population-based study of eclampsia: lessons learnt to improve maternity care

PONE-D-23-10835R1

Dear Dr. Korb

We’re pleased to inform you that your manuscript has been judged scientifically suitable for publication and will be formally accepted for publication once it meets all outstanding technical requirements.

Kind regards,

Natasha L Pritchard

Academic Editor

PLOS ONE

Additional Editor Comments (optional):

Dear Dr Korb

Thank you for the response. They adequately address the points raised, and your manuscript would add value in the literature.

Kind regards